

# A High-Resolution Global SWAT+ Hydrological Model for Impact Studies.

Celray James Chawanda[1,2], Ann van Griensven[1,3], Albert Nkwasa[1,4], Jose Pablo Teran Orsini[1], Jaehak Jeong[2], Soon-Kun Choi[5], Raghavan Srinivasan[2], Jeffrey G. Arnold[6],

[1] Department of Water and Climate, Vrije Universiteit Brussel, Elsene, 1050, Belgium
[2] Blackland Research & Extension Center, Texas A&M Agrilife Research, Temple, 76502 TX, USA
[3] Institute for Water Education (IHE) Delft, 2611 AX Delft, The Netherlands
[4] International Institute for Applied Systems Analysis (IIASA), A-2361 Laxenburg, Austria
[5] National Institute of Agricultural Sciences (NAS), Rural Development Administration, Republic of Korea
[6] Grassland Soil and Water Research Laboratory, USDA Agricultural Research Service (ARS), Temple, 76502 TX, USA

*Correspondence to*: Celray James Chawanda (celray.chawanda@tamu.edu)

## Abstract

Global hydrological models are essential tools for understanding water resources and assessing climate change impacts at planetary scales, supporting water management, flood risk assessment, and sustainable development initiatives worldwide. The Soil and Water Assessment Tool (SWAT+) has demonstrated robust performance across various environments and scales, from local to continental applications. However, despite its widespread use, a global implementation of SWAT+ is currently lacking due to computational demands and data management challenges, while existing global models often lack the detailed process representation and high spatial resolution needed for comprehensive hydrological analysis. A global SWAT+ model would offer unique advantages through its integrated simulation of water quantity, quality, and land management processes, while supporting multiple UN Sustainable Development Goals and enhancing research opportunities in global hydrology. This study aimed to develop a High-resolution Global SWAT+ Model and establish a reproducible framework for large-scale SWAT+ applications. We developed the Community SWAT (CoSWAT) modeling framework, an open-source solution that automates data retrieval, preprocessing, and model configuration using Python, while maxmising parallel processing for computational efficiency. The global model was then set up using the framework at 2km resolution using ASTER DEM, ESA land use data, FAO soil data, and ISIMIP climate data, with performance evaluated against GRDC flow data and GLEAM evapotranspiration dataset. Results without calibration showed reasonable spatial patterns in evapotranspiration simulation with 78.54% of sampled points showing differences within ±100mm compared to GLEAM data, though river discharge performance was limited due to lack of reservoir implementation with 23.02% of stations showing positive Kling-Gupta Efficiency values. The development of this first global SWAT+ model demonstrates the feasibility of high-resolution global hydrological modeling using SWAT+, while the CoSWAT framework provides a robust





foundation for reproducible large-scale modeling. These advances enable more detailed analysis of global water resources and climate change impacts, though future work should focus on incorporating water management practices, improving process representation with calibration, and enhancing computational efficiency.

**1. Introduction**

Global water models aim to simulate phenomena at planetary scale. There are several global models that simulate the water cycle such as Global Hydrological Models (GHMs), Land Surface Models (LSMs), and Global Dynamic Vegetation Models (GDVMs), each of them with different approaches and purposes. Most models are based on a gridded structure and vary in how they represent water storage components and terrestrial processes. Nearly all models incorporate canopy, snow, and soil

water storage to some degree. Surface runoff is estimated by most models, while only a smaller group—specifically GHMs and a number of LSMs—include river routing to enable streamflow simulations. Additionally, only a small fraction of models account for reservoirs, with an even smaller subset incorporating lake and wetland storage (Telteu et al., 2021).

In order to assess water resources on a global scale, multi-model comparisons exist due to the differences in structures and

approaches of global water models. There are some  Model Intercomparison Projects (MIPs) that focus on global water models such as the WaterMIP (Haddeland et al., 2011), and the Inter-Sectoral Impact Model Inter-Comparison Project (ISIMIP) that, on the ISIMIP 2 and 3 simulation rounds(Gosling et al., 2023a, b, 2024a, b), involves several models including GHMs, LSMs and DGVMs, as shown in Table 1.

**Table 1: Global Water Models utilized in the global water sector of ISIMIP**

| Nr | Name | Category | Reference |
|---|---|---|---|
| 1 | CWatM | *GHM* | *(Burek et al., 2020)* |
| 2 | DBH | *GHM* | *(Tang et al., 2007)* |
| 3 | H08 | *GHM* | ***(Hanasaki et al., 2018a)*** |
| 4 | HydroPy | *GHM* | *(Stacke and Hagemann, 2021)* |
| 5 | MPI-HM | *GHM* | *(Stacke and Hagemann, 2012)* |
| 6 | PCR-GLOBWB | *GHM* | *(Sutanudjaja et al., 2018)* |
| 7 | VIC | *GHM* | *(Liang et al., 1994)* |
| 8 | WaterGAP2 | *GHM* | *(Müller Schmied et al., 2021)* |
| 9 | WAYS | *GHM* | *(Mao and Liu, 2019)* |
| 10 | WEB-DHM-SG | *GHM/LSM* | *(Qi et al., 2022c)* |
| 11 | CLASSIC | *LSM* | *(Melton et al., 2020)* |
| 12 | CLM 4.5 | *LSM* | *(Oleson et al., 2013)* |
| 13 | CLM 5.0 | *LSM* | *(Lawrence et al., 2019)* |
| 14 | ELM-ECA | *LSM* | *(Zhu et al., 2019)* |
| 15 | JULES | *LSM* | *(Best et al., 2011)* |
| 16 | MATSIRO | *LSM* | *(Takata et al., 2003)* |
| 17 | ORCHIDEE | *LSM* | *(Guimberteau et al., 2014, 2017)* |



| 18 | SWBM | LSM | (Rene Orth and Seneviratne, 2015) |
| 19 | SSiB4/TRIFFID | LSM/DGVM | (Huang et al., 2020) |
| 20 | LPJmL | DGVM | (Sitch et al., 2003) |

Among these models, generally, GHMs focus on representing the land water balance, with particular emphasis on streamflow for model evaluation, while LSMs focus on the vertical exchange of energy and water between land and atmosphere, and DGVMs simulate global vegetation dynamics, where hydrological processes play an important role (Telteu et al., 2021).

The Community Water Model (CWATM), is a gridded rainfall-runoff and channel routing hydrological model developed for assessing water quantity on a global scale, considering water consumption across multiple sectors (Burek et al., 2020), and it has been applied to studies about energy, water security (Palazzo et al., 2024) and water supply (Becher et al., 2024). The Distributed Biosphere Hydrological model (DBH) combines a hydrological scheme with a biosphere model, able to estimate surface runoff and river routing. It consider human influences such as irrigation (Tang et al., 2007), and it has been used to assess regional impacts of climate change on the water cycle (Tang et al., 2008).

The H08 model simulates surface runoff, river routing, reservoir operation, crop growth and water abstractions. It was developed to map water abstractions from different sources of water and study global water availability (Hanasaki et al., 2018a), as well as estimating water scarcity indicators on a global scale (Hanasaki et al., 2018b). It has been used for studying future impacts of CC on global irrigation water withdrawals (Haile et al., 2024), and for assessing future influence of regulation structures on flood risk (Boulange et al., 2021). The Max Planck Institute – Hydrology Model (MPI-HM) performs global simulations of surface water balance and river streamflow, with application in wetland dynamics (Stacke and Hagemann, 2012). HydroPy is a revised version of the model, fully written in Python (Stacke and Hagemann, 2021).

PCRaster Global Water Balance (PCR-GLOBWB) simulates the terrestrial water cycle, considering multiple storages such as canopy, soil, snow, floodplains, lakes and rivers. It has been used to simulate human influences on the water cycle (Sutanudjaja et al., 2018) and for studies on global water resources (Hoch et al., 2023; Long et al., 2015) and flood risks (Hoch et al., 2023). The Variable Infiltration Capacity (VIC) is a gridded land surface hydrology model, originally part of the Geophysical Fluid Dynamics Laboratory (GFDL) global circulation model (Liang et al., 1994), able to perform river routing, and simulate lake and wetland water balance. It has been used to study human impacts on global water resources (Droppers et al., 2020; Hamman et al., 2018).

The Water Global Assessment and Prognosis (WaterGAP) simulates surface runoff, groundwater recharge and river streamflow, considering storages such as snow, soil, aquifers, lakes, wetlands and rivers (Müller Schmied et al., 2021) and has been used for studies related to global water availability and use (Müller Schmied et al., 2016; Wan et al., 2024), and for



estimating drought indices on a global scale (Herbert and Döll, 2023) . The WAYS model is a distributed hydrological model, which was developed focusing on representing root zone water storage, and simulates soil water dynamics adequately representing field-scale heterogeneity of the soil (Mao and Liu, 2019).

The Water and Energy Budget-based Distributed biosphere Hydrological Model with improved Snow physics for Global simulation (WEB-DHM-SG) is a combination of an LSM and a GHM, able to simulate the land surface energy, water balance and river routing. It has been used for regional studies assessing the changes in land surface properties and global warming on water resources(Qi et al., 2019b) and the contribution of snow to the water availability of several river basins(Qi et al., 2020, 2022b).

Models such as the Community Land Model (CLM) 4.5 and CLM 5.0 were developed for the Community Earth System Model (CESM), and their simulations include biogeophysical, hydrological, biogeochemical and vegetation processes of the land. The Canadian Land Surface Scheme Including Biogeochemical Cycles (CLASSIC) is the result of a coupling of the Canadian Land Surface Scheme (CLASS) and the Canadian Terrestrial Ecosystem Model (CTEM), which simulates the energy and water cycles, as well as biogeochemical cycles (Melton et al., 2020). The land model of the Energy Exascale Earth System Model (E3SM) is the ELM-ECA land model, which was developed mainly to simulate biogeochemical interactions (Zhu et al., 2019). The Joint UK Land Environment Simulator (JULES) simulates energy, water, carbon fluxes and vegetation dynamics, it has several versions for which different process calculation schemes (e.g., river routing) have been used (Best et al., 2011; Clark et al., 2011). Models like the Minimal Advanced Treatments of Surface Interaction and Runoff (MATSIRO), ORCHIDEE, SWBM, and the Simplified Simple Biosphere Model coupled with the Top-down Representation of Interactive Foliage and Flora Including Dynamics Mode (SSiB4/TRIFFID), similarly to the LSMs previously mentioned, are standalone land surface models that simulate the water cycle and biogeochemical cycles. The Lund-Potsdam-Jena managed Land (LPJmL) is a multisectoral DGVM which simulates the surface water balance, and includes human influences such as irrigation. It has been used to simulate the impacts of CC on water availability (Drüke et al., 2021; Schaphoff et al., 2018).

Despite the differences in general purposes and focus for model development with between LSMs, GHMs and DGVMs, as part as the model ensemble for the global water sector from ISIMIP (Table 1), they have been used together for several studies.  On a global scale, model ensembles were used to study the future changes in groundwater recharge (Reinecke et al., 2021). They were applied to assess the historical and future impacts of Climate Change (CC) on river flow trend and soil moisture(Gudmundsson et al., 2021; Porkka et al., 2024; Thompson et al., 2021). Moreover, others have studied historical and future changes in drought and flood risks, trends and impacts (Dottori et al., 2018; Kew et al., 2021; Pokhrel et al., 2021; Tabari et al., 2021; Zhou et al., 2023). On a regional scale, ensembles of models were used to study compound extreme





climate events (Muheki et al., 2024), and project future indices regarding water scarcity in the context of CC and societal changes (Yin et al., 2020).

The ability to simulate water resources and the impacts of CC and other human induced environmental changes is
indispensable in planning for and management of water resources (Ramteke et al., 2020; Soltani et al., 2023; Zhuang et al., 2018). This is true for both small scales and large scales (Chawanda et al., 2024; Fu et al., 2019). By predicting how CC and other human drivers affect the water cycle, the models help in developing strategies to cope with droughts, floods and other water-related challenges (Brunner et al., 2021). Hydrological models also enable more effective management of water resources to optimize the use of water for agriculture (Li et al., 2020; Srivastava et al., 2020), industry, and human
consumption, especially in regions where water is scarce (Hanasaki et al., 2018b). These models at a global level can also support global sustainable initiatives by ensuring that development projects align with long-term water availability (Amjath-Babu et al., 2019). At the same time, they can also offer means for forecasting extreme events like floods and droughts in any region of the world, thereby enhancing preparedness and allowing timely warning for disaster response efforts.

SWAT+ (Soil and Water Assessment Tool) is a completely revised version of the original SWAT model (Arnold et al., 2018; Bieger et al., 2017). The SWAT+ model can simulate a wide range of processes including surface runoff and infiltration, evapotranspiration and other water balance components (Pandi et al., 2023). SWAT+ also simulates Soil Erosion and Sediment Transport, Nutrient Cycle and Land Use and Management Practices (Arnold et al., 2018). SWAT has been applied all over the world in various environments including in Temperate (Qi et al., 2019a), Tropical and Subtropical (Alemayehu et
al., 2017; Ma et al., 2019), Arid and Semi-Arid (Samimi et al., 2020), Mediterranean Climates, Cold and Mountainous Regions, Wetlands and even Coastal Environments (Peker and Sorman, 2021; Pulighe et al., 2021; Upadhyay et al., 2022). SWAT+ has also been applied at small scales (Qi et al., 2022a), regional scale (Chawanda et al., 2020a; Nkwasa et al., 2022b) and even continental scale (Abbaspour et al., 2015; Chawanda et al., 2024; Nkwasa et al., 2024)

Despite such applications, there are no global applications at present because as with other global modelling efforts, large-scale SWAT+ applications face several significant challenges. Data availability and quality remain a challenge (Crochemore et al., 2020). High-resolution, consistent, and up-to-date datasets for land use, soil properties, and climate variables are often lacking or incomplete (Chawanda et al., 2024; Döll et al., 2016), particularly in developing regions which can lead to increased uncertainty (Sood & Smakhtin, 2015) in model outputs (Sood & Smakhtin, 2015) and limit the model's
applicability. Computational demands pose another challenge (Ma et al., 2023; Zhang et al., 2016). The computational requirements of setting up fine resolution SWAT+ model, running, calibrating and validating it, coupled with the storage resources required for the input and output data, necessitates significant computational resources. These two in addition to data processing methods used in global applications, can make it challenging to replicate and reuse any model set-ups available in a study area (Chawanda et al., 2020b). This issue may be compounded by frequent updates to model structure



and parameters (Smith et al., 2020), which can lead to inconsistencies between studies conducted at different times. This calls for archiving and versioning systems in workflows for better reproducibility (Knoben et al., 2022).

While challenges persist, a global SWAT+ model is in a unique position to provide comprehensive insights into large-scale processes across diverse ecosystems worldwide. To begin with, a global SWAT+ model offers a holistic approach in

simulation of water quantity and quality, integrating detailed hydrological processes, nutrient cycling, and sediment transport (Abbaspour et al., 2015; Liu et al., 2017). The model also would enable high-resolution projections of climate change impacts on global water systems, critical for contributing to international assessments like Intergovernmental Panel on Climate Change (IPCC) reports. In addition, the global SWAT+ model directly supports multiple United Nations Sustainable Development Goals (SDGs), by providing data crucial for balancing development with environmental conservation through

simulation of Land Use and Land Cover Change (LULCC) and nutrients in water bodies. Implementing a global SWAT+ model also creates a standardised dataset for cutting-edge international research in hydrology, thereby enhancing research and educational opportunities. The model has significant implications in designing and enhancing the effectiveness of long-term global water and land management practices by allowing detailed simulations of global agricultural practices, land use changes, and their water resource impacts.


The primary aim of this study was to develop a High-resolution Global SWAT+ Model, addressing the growing need for comprehensive, large-scale hydrological simulations. To achieve this overarching goal, we first establish a robust framework for setting up a high-resolution global SWAT+ model based on the study by Chawanda et. al., (2020b), ensuring reproducibility and scalability of the model. This framework integrates global data processing methods and computational

strategies that overcome the challenges inherent in global-scale modelling. Subsequently, we evaluate the model's performance against other established global models and observed data. This evaluation not only benchmarks the Global SWAT+ Model's capabilities, but also identifies areas for potential refinement, contributing to the advancement of global hydrological modelling techniques.



## 2. Methodology

### 2.1 Global Datasets for SWAT+

While setting up the global SWAT+ model, global data sources were used as input and for model evaluation.

### 2.1.1 Digital Elevation Model

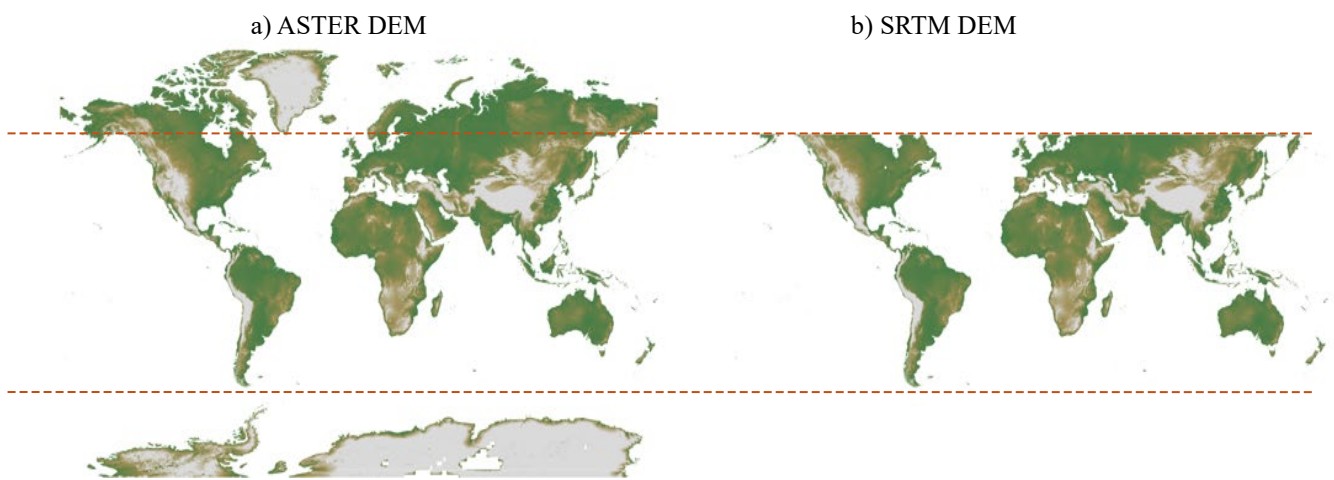

**Figure 1: Spatial coverage of ASTER Global DEM vs SRTM. SRTM data only cover over 80% of the Earth's land surface (60°N–56°S), while ASTER goes further North and South (Yue et al., 2017).**

The Advanced Spaceborne Thermal Emission and Reflection Radiometer (ASTER) global DEM (Abrams, 2016) was preferred over the Shuttle Radar Topography Mission (SRTM) global DEM (Farr et al., 2007) due to spatial coverage (Fig. 1).

### 2.1.2 Land Use Map

The land use data (Fig. 2) from European Space Agency (ESA) was downloaded from the ESA website (ESA. Land Cover CCI Product User Guide Version 2. Tech. Rep. (2017). Available at: maps.elie.ucl.ac.be/CCI/viewer/download/ESACCI-LC-Ph2-PUGv2_2.0.pdf)



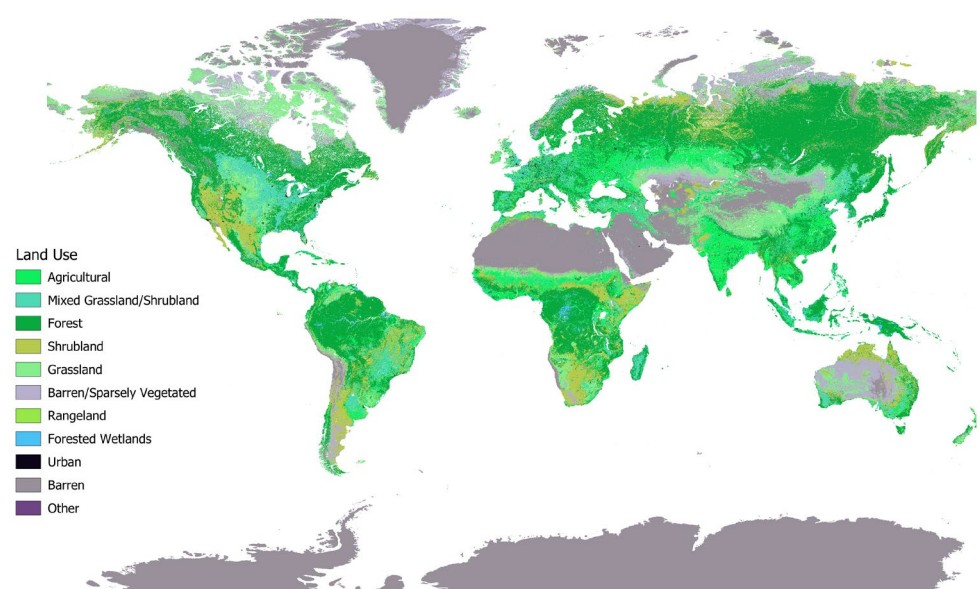

**Figure 2: Major Land Use Categories from European Space Agency (ESA) Land Use Map**

### 2.1.3 Soil Map

The FAO soil data (Fischer et al., 2008) was used in this study. The FAO soil data, particularly from the Harmonized World

Soil Database (HWSD), provides global coverage of soil properties (Fig. 3) at a 1 km² resolution. The data is derived from
multiple sources and is widely used in global and regional environmental studies, though it may lack precision for local-scale
analysis due to the generalised nature of its source material.

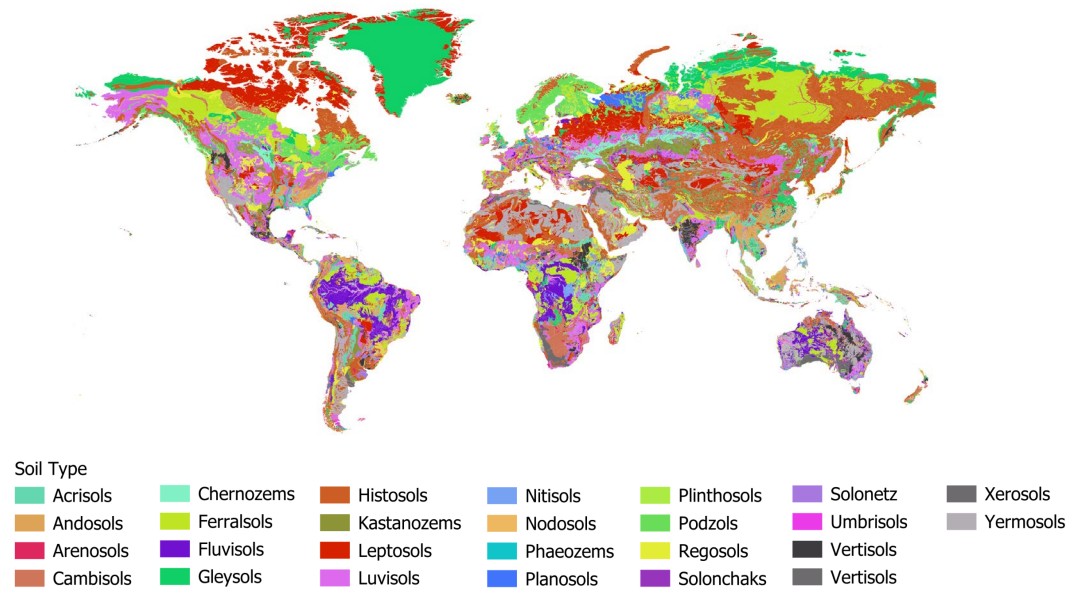

**Figure 3: Major soil types from FAO Soil Map**



### 2.1.4 Climate Data

Climate data was acquired from the GSWP3-EWEMBI reanalysis dataset (Lange and Büchner, 2020) through the Inter-Sectoral Impact Model Intercomparison (ISIMIP) project. The dataset contains historical climate data including daily minimum and maximum air temperature (tasmin and tasmax, respectively), precipitation (pr), relative humidity (rhs), solar radiation (rlds), and near-surface wind speed (wind) at 0.5 decimal degrees.

### 2.1.5 River Flow Data

The monthly river flow for evaluation was acquired from The Global Runoff Data Centre (*GRDC*, https://portal.grdc.bafg.de/applications/public.htm). The gauging station location data was also used in the delineation to create outlets in the model setups.

### 2.1.6 ET Data

Gleam4 dataset was used for evaluating ET (Miralles et al., 2011, 2024) The datasets require preprocessing to be used by the SWAT+ model. In our model set up, the preprocessing was done in the scripted workflow described in the next section.

### 2.2 Scripted Workflow for Global SWAT+ Setup

To facilitate the setup of a global SWAT+ model, we developed a python based scripted workflow based on SWAT+ AW (Chawanda et al., 2020b). This new workflow, named the Community SWAT (CoSWAT) modelling framework, is a free and open-source solution designed for large-scale modelling using SWAT+. The framework simplifies the setup process for global SWAT+ models by automating data retrieval, pre-processing, and model configuration.



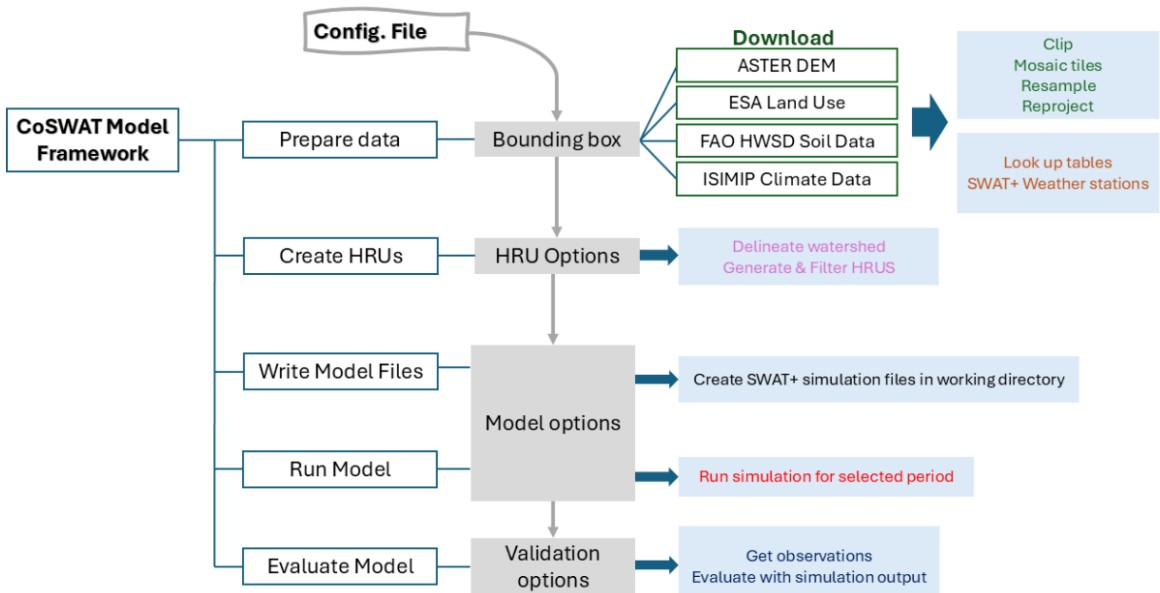

**Figure 4: Schematic**

The user is required to define regions of interest using box coordinates. Settings for model set up are saved in one file referred to as the 'configuration file'. Having model setup settings in one file ensures consistency across the global model. To ensure reproducibility and ease of use, the workflow includes stages for retrieving and preparing essential input datasets. These stages are automated and controlled by the configuration file settings:

- Digital Elevation Model (DEM): The CoSWAT framework downloads ASTER DEM tiles, mosaics them into a
continuous surface, and resamples and reprojects them as per the configuration file.

- Land Use Data: The framework automatically retrieves land use maps from the European Space Agency (ESA) and processes them. The data is resampled, and reprojected. The lookup file is manually prepared once. This is done by matching ESA land use classification to equivalent land use types in the SWAT+ land use database.

- Soil Data: Soil data from the FAO Harmonized World Soil Database (HWSD) is downloaded and transformed into the
format required by SWAT+. The framework handles rasterization and reprojection to prepare the soil data into a format required by SWAT+. FAO soil lookup and soil properties database are readily available from the SWAT+ website and are used by the workflow.

- Climate Data: Climate inputs are downloaded from the ISIMIP dataset servers. The workflow processes historical and scenario-based climate data, including variables such as precipitation, temperature, wind speed, solar radiation and
relative humidity and formats them for use in SWAT+ simulations. The downloaded files are in NetCDF format and are read and written using the xarray python library. Users can customise climate scenarios using the configuration file. It is also possible to specify the spatial resolution of the points at which climate time series are created.



The flexibility of the CoSWAT Framework allows users to easily switch data sources, depending on the project's
requirements or data availability which makes the framework a robust and adaptable tool for large-scale modelling.

In addition to the model setup, the CoSWAT framework integrates evaluation and visualisation tools. A local web application
was developed to serve as an interactive portal, allowing users to visualise model results and outputs in a user-friendly
environment. This platform enhances the accessibility and interpretation of model results. The CoSWAT framework was
optimised by iteratively implementing parallel processing wherever possible and feasible. This reduces the time required for
data processing and model setup, making large-scale simulations feasible by leveraging High Performance Computing
(HPC) environments which often allow highly parallelised workflows (Chawanda et al., 2020b).

**Global SWAT+ Model Implementation and Evaluation**

The data resolution for DEM, land use and soil maps were set to 2km projected in ESRI:54003 (Miller World Cylindrical)
projection. Climate data resolution was however set to 0.5 decimal degrees due to limitations on the number of files the
operating system allowed (<= 10,000,000 files). The thresholds for stream and channel were set to 44 cells – an equivalent of
177.7 km$^2$.

The model was setup in a 64-core HPC Environment running at 3.00 GHz, with 128 GB Memory running Linux. To take
advantage of parallel processing and easy data handling, the global model was setup by combining regions defined based on
major river basins (Fig. 5). However, Greenland was not included in this version.

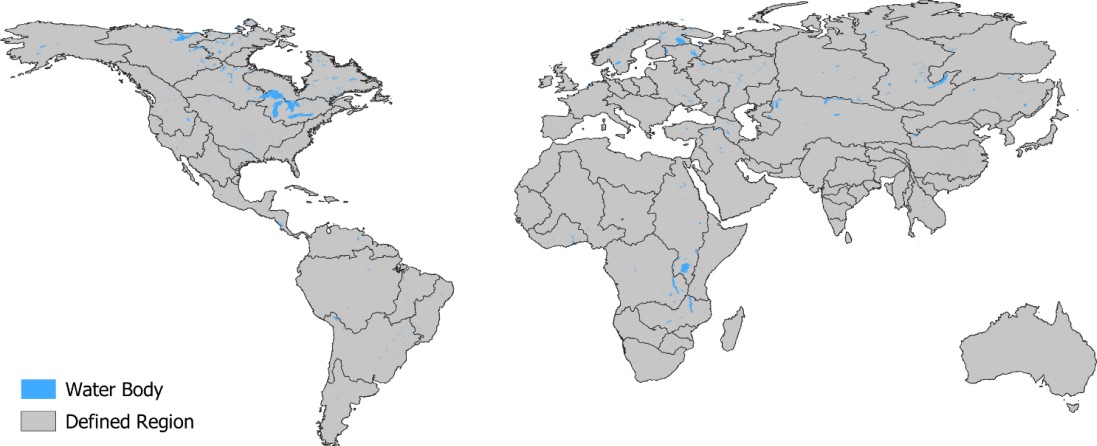

**Figure 5: Partitioning of the world land mass into regions based on main river basins.**

Using the CoSWAT framework, data was prepared, and a model set up for each of these regions using the configuration file
options. Slope classes were not considered in creation of HRUs.  The Model Files were written with the following options



**Table 2: Model configuration options used when writing model files.**

| Routing Method | Muskingum |
|---|---|
| PET Estimation Method | Penman-Monteith |
| Warm up Period | 5 years |
| Simulation Period | 1977 - 1990 |

The version of SWAT+ used to run the global SWAT+ model setup is 60.5.7. The global model was not calibrated as that was beyond the scope of this study. We used the Kling-Gupta Efficiency (KGE) metric (Gupta et al., 2009) to evaluate the flow time series at the monthly timestep. The plotting of results and metric calculation was implemented within the CoSWAT
framework.

The Kling-Gupta Efficiency (KGE) metric provides a more nuanced assessment of model performance than conventional measures like R² or Bias alone. It is formulated to simultaneously account for three key components: the linear correlation between simulated and observed time series (r), the variability ratio – α, which compares the standard deviation of simulated values against observations, and the bias ratio – β, which evaluates the mean offset between simulated and observed values
(Gupta et al., 2009). KGE gives a balanced indication of how well the model reproduces both the overall magnitude and the temporal dynamics of observed flows. By considering these complementary aspects.

In addition to reporting the KGE values, we also present the underlying distributions of r, α, and β across all gauging stations. This helps to reveal a global picture of how the model simulations compare against observed data.

We also evaluated the ET output against GLEAM v3 dataset using maps and sample point difference distribution. Sampled
points were quasi-randomly selected in a way to reduce clustering (Fig. 6).

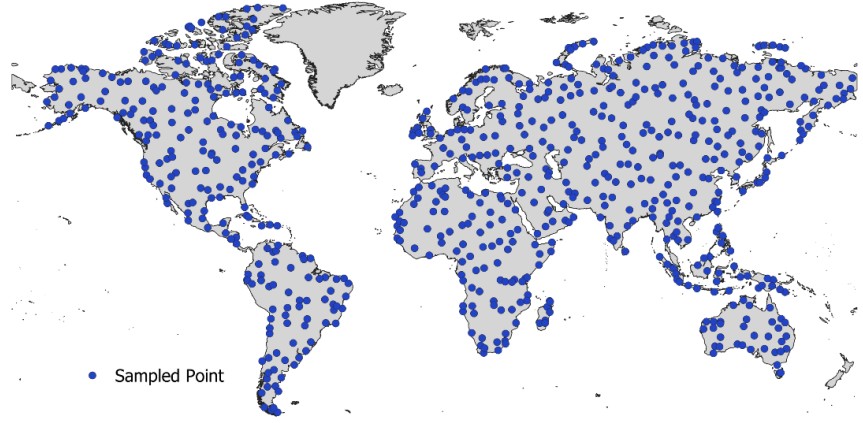

**Figure 6: Sampled points for measuring distribution of ET differences between Gleam and SWAT+ output.**





# 3. Results

In total, there were 2.63 million HRUs. Figure 7 illustrates the level of discretisation that was achieved in creating HRUs.

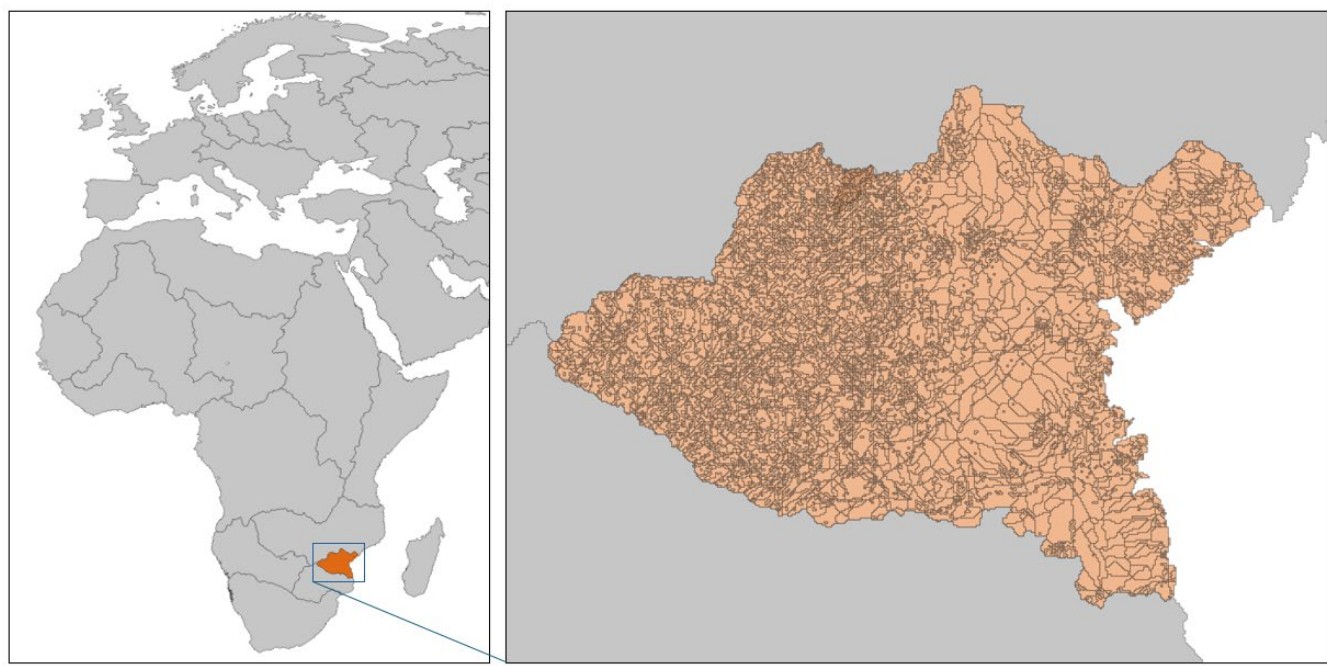


**Figure 7: Illustration of density of HRUs in the Save Region of the African Continent**

When loaded into the Django visualisation app (Fig. 8), the Web User Interface (UI) showed all gauging stations where the user can pan around and click on any station to see details including performance metrics.

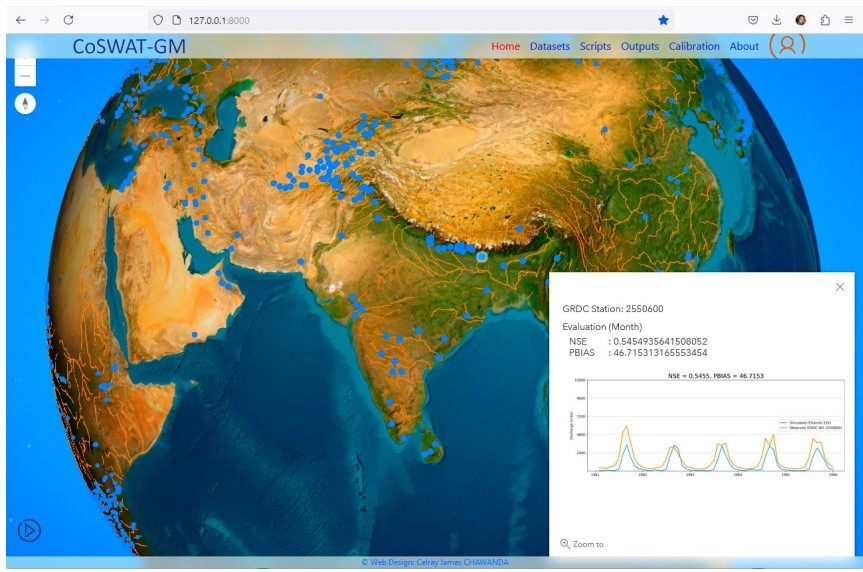

**Figure 8: Django Web User Interface for visualising and managing the CoSWAT Global Model (https://github.com/celray/coswat)**



The web UI also allows users to see which datasets were used, download extract outputs from the model and acts as a calibration portal. Thus, users can extract a region to calibrate and update input files for better model performance.

## 3.1 Evapotranspiration

When compared with Gleam ET, the spatial pattern between SWAT+ ET and GLEAM ET is comparable overall.


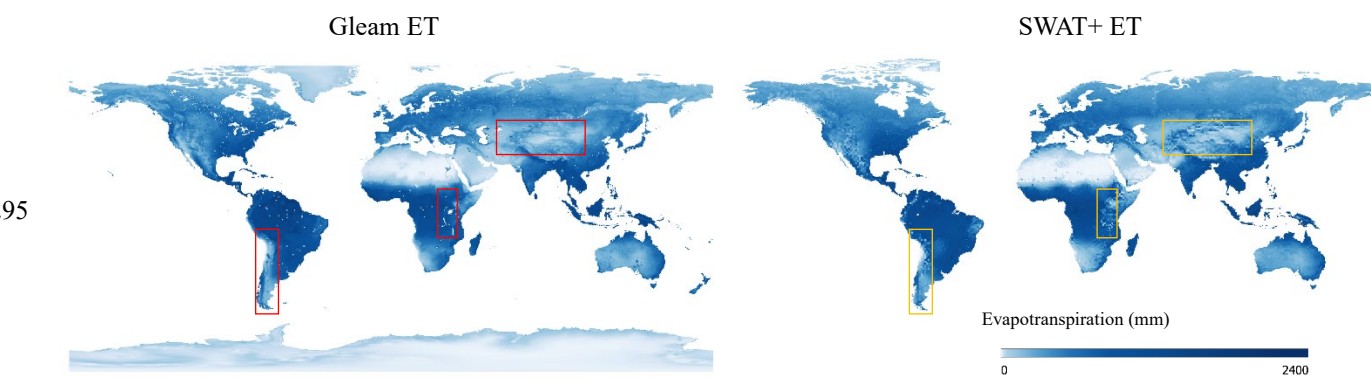

**Figure 9: A comparison of the Gleam and SWAT+ ET maps. Highlighted regions show artefacts in the SWAT+ Output due to lack of representation of large water bodies and snow issues in high mountains.**

However, SWAT+ ET appears to have artefacts on the ET in East Africa and Central Asia. It also fails to capture the pattern in South America along the Andes mountains. Figure 10 also shows how sampled points compare in differences with 78.54% within the range -100mm to 100mm ET difference.

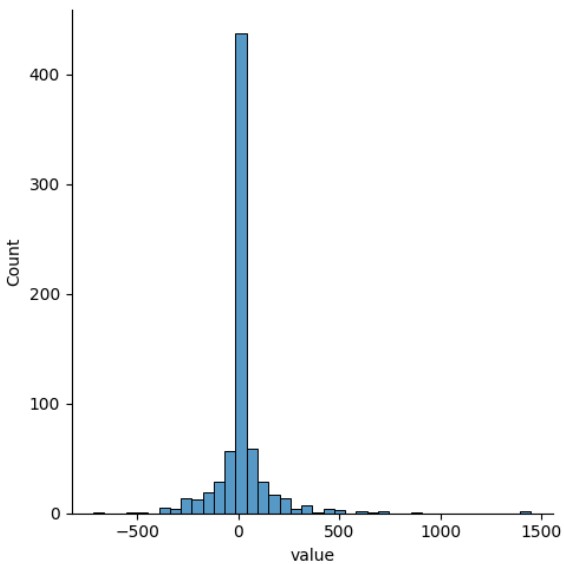

**Figure 10: Distribution of map difference ET values (GLEAM – Model ET)**





## 3.2 River Discharge

The model achieved some positive KGE values (23.02%), but a majority of the values were negative as demonstrated by Fig. 11.

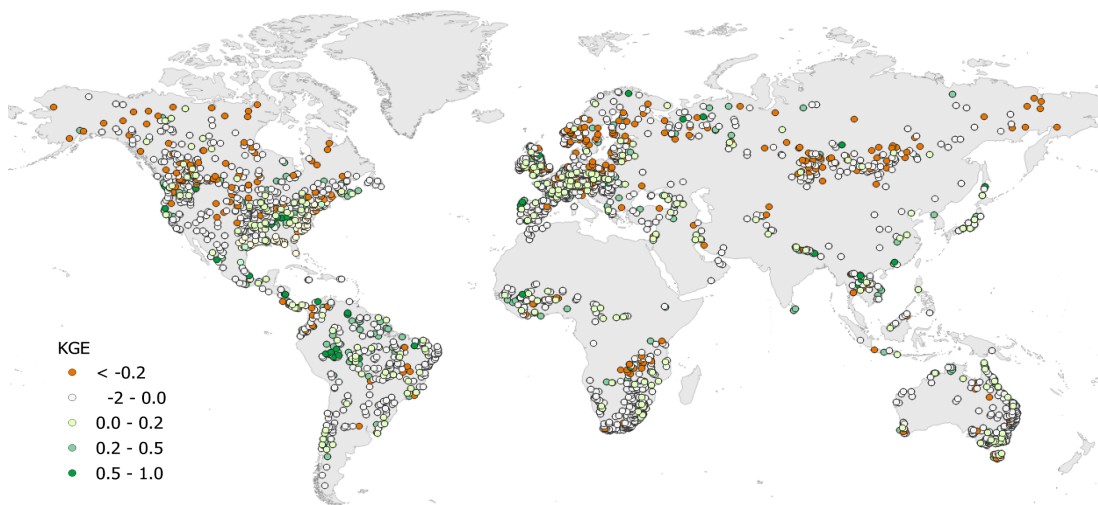

**Figure 11: River Flow Performance (KGE)**

While a small percentage of KGE values were above 0, 85.31% of the stations showed a positive corelation (r) in flow values with variability ratio (α) and mean ratio (β) median values falling close to 1.

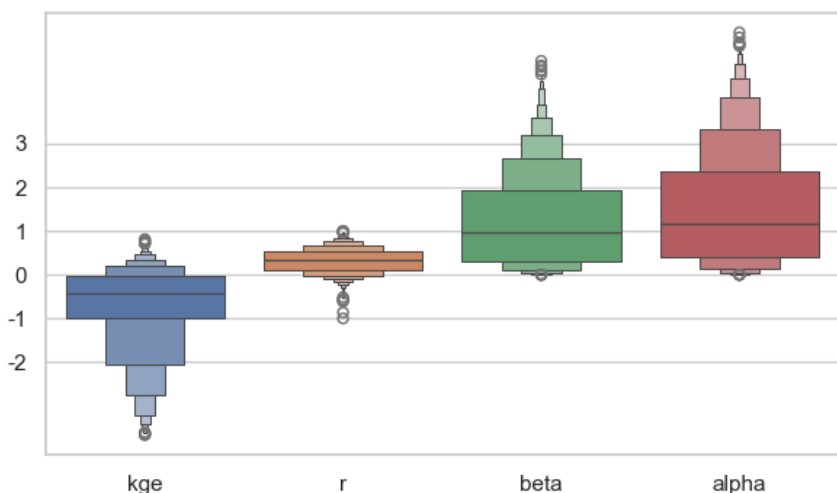

**Figure 12: Boxen Plots showing the distribution of kge, r, alpha (α) and (β)**



## 4. Discussion and Future Work

The development of the global SWAT+ model using the CoSWAT framework highlights several significant advancements in large-scale hydrological modelling using SWAT+. The scripted workflow approach offers key advantages. By automating data retrieval, preprocessing, and model configuration within a single framework, the CoSWAT workflow ensures that the model setup can be consistently replicated. This is crucial for verifying results and facilitating collaborative research efforts. The availability of the workflow as an open-source tool further enhances transparency and community engagement. The CoSWAT framework also efficiently leverages multicore processing capabilities of high-performance computing (HPC) environments which reduces computational time, making it practical to perform high-resolution global simulations that were previously computationally prohibitive (Chawanda et al., 2020b). Users can easily adjust the model setup by modifying the configuration file, allowing for easy updates to input data sources, spatial resolution, or simulation parameters. This flexibility is essential for adapting the model to different research questions or incorporating new datasets as they become available.

With the built in Django App for visualization, the CoSWAT framework plays an important role in visualising results. Users can zoom in to given sections of the model setup to extract and improve model setup separately.

The first version of the CoSWAT global model showed promising results despite not being calibrated. The global SWAT+ model demonstrated reasonable performance in simulating ET when compared with the GLEAM dataset. The spatial patterns were generally consistent, with approximately 78.54% of sampled points showing differences within ±100 mm. However, discrepancies were noted in regions such as East Africa, Central Asia, and along the Andes mountains in South America. These artefacts may be due to input data quality. Limitations in the resolution or accuracy of input datasets, particularly climate data, can affect ET simulation. Reanalysis datasets may not capture local climate variability effectively, especially in regions with complex topography or sparse observational data (Moalafhi et al., 2017). The absence of lake representation in the SWAT+ Model setup also contributed to some of the discrepancies. For instance, the East African rift valley lake area was all simulated with regular HRUs while implementing lakes would ensure that the land ET and lake ET are not mixed up to improve spatial Pattern (Fig 9).

The model's performance in simulating river discharge was less satisfactory, with only 23.02% of gauging stations showing positive Kling-Gupta Efficiency (KGE) values for the simulation period (1977 – 1990 with 5 year warm up). Several factors contributed to this outcome. Without calibration, the model relies on default parameters, which may not reflect the hydrological characteristics of diverse global regions. Calibration would help better represent hydrological processes by adjusting model parameters to match observed streamflow and improve performance (Molina-Navarro et al., 2017). However, at such a scale, calibration would not be easily feasible due to computational and data storage requirements. Chawanda et al 2020 detail how Hydrological Mass Balance Calibration (HMBC) applied at large scale improves water balance representation while also improving model performance in several gauging stations in at a feasible computational



cost. This is demonstrated by their application of HMBC on the SWAT+ Model for Africa (Chawanda 2024). Global modelling exercises like CoSWAT global model can also incorporate such calibration routines to improve the model
performance.

The non-inclusion of reservoirs and water management practices also negatively affected the performance of the model. The model did not implement reservoirs or account for human interventions such as irrigation, which significantly impact river flow regimes. Large-scale hydrological models struggle with representing human activities accurately due to their
complexity and data requirements (Wada et al., 2017). Including reservoirs and management practices is an important part for realistic flow simulations, as demonstrated by Chawanda et al. (2020a), who showed that incorporating these elements improves river flow and ET simulations.

Poor performance was also noted in higher latitudes which may be attributed to excessive simulation of snow, leading to
overestimated flows. This suggests a need to refine the model's snow routines or adjust parameters related to cold climate processes. While snow related performance issues are specific to high latitude areas, the use of reanalysis climate data at a 0.5-degree likely caused reduced model performance throughout the model setup. The 0.5-degree resolution may not capture local-scale climate variations, particularly in regions with significant topographic variability (Kay et al., 2015). Downscaling techniques or higher-resolution datasets could enhance model performance (Wang et al., 2020; Zhu et al., 2023). One major
limitation faced during the simulation of the global model, the current SWAT+ climate input system requires individual files for each variable at each weather station, resulting in a massive number of files for global models. In our Model Setup, we required about 230,000 climate files at 0.5 decimal degree resolution. This approach strains computational resources, slows disk access, and increases memory usage, especially for long-term simulations or multiple scenarios and limits applicability of downscaling efforts. There is a need to modify the input system to handle large datasets more efficiently by adopting and
integrating more efficient file formats like NetCDF which would only need one file for all timesteps and climate variables.

Future model versions should include reservoirs, irrigation, and other water management practices to capture both natural and anthropogenic activities. This requires collection of global datasets on water infrastructure and usage, which can be challenging but is essential for better process representation (Nkwasa et al., 2022a). In addition, HMBC should be employed
to further improve process representation and hence model performance. Snow process and parameters also need to be revised to prevent snow build up in higher latitudes.

## 5. Conclusions

The development of a high-resolution global SWAT+ model using the CoSWAT framework marks a significant step forward in global hydrological modelling. The framework's reproducibility, scalability, and flexibility address many challenges



associated with large-scale simulations. While the model performed well in simulating evapotranspiration, discrepancies in certain regions highlight the need for further refinement of input data and model parameters.

The poor performance in river discharge simulations highlights the importance of model calibration and the inclusion of human activities such as reservoir operations and water management practices. Future work should focus on enhancing the representation of critical hydrological processes, integrating human interventions, and improving input data quality and

resolution. Adopting more efficient data handling strategies within the SWAT+ framework will also facilitate larger and more complex simulations. By addressing these challenges, the global SWAT+ model can become a powerful tool for understanding global water resources and easily map hotspots for water scarcity, assessing climate change and land use impacts, and supporting sustainable water management practices worldwide.

**Code and data availability**

Simulations have a large size and cannot easily be hosted online. However, they are available upon request. The tools used in this study are available from github (https://github.com/celray/CoSWAT-Framework and https://github.com/celray/coswat, last access: 28 December 2024) and through Zenodo at https://zenodo.org/doi/10.5281/zenodo.14577842 (Chawanda, 2024). All input data is from open sources as discussed in the manuscript.

**Aknowledgements**

The authors thank the Research Foundation – Flanders (FWO) for funding the International Coordination Action (ICA) "Open Water Network: Open Data and Software tools for water resources management" (project code G0E2621N), the Open Water Network: impacts of global change on water quality (project code G0ADS24N), the AXA Research Chair fund on Water Quality and Global change and the King Baudouin Foundation for the Ernest du Bois Prize Fund (agreement No. 2022-F2812650-228938).


**Author contributions** CJC conceptualized the study, performed the investigation and analysis. All authors contributed to the result interpretation and review. CJC wrote the first draft of the manuscript, and all authors commented and revised the manuscript. All authors read and approved the final manuscript.

**Competing interests**

The authors declare that they have no conflict of interest.





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
