# Peer review of "A High-Resolution Global SWAT+ Hydrological Model for Impact Studies."

_EGUsphere, 2025_

## Author Comment (AC1)

**Responses to Reviewers**

Purple is quoted comment; Orange is a response, and Blue is quoted text from manuscript.

**Reviewer 1**

- In the introduction, Global Hydrological Models (GHMs) are described in several separate paragraphs, which makes the information somewhat fragmented. I recommend consolidating this into a more concise summary, perhaps in two or three paragraphs. This will help to present the key aspects of the model more clearly and cohesively, improving the overall readability of the section.

We thank the reviewer for the constructive feedback regarding the description of Global Hydrological Models (GHMs) in the introduction. We agree that consolidating the information will improve clarity and cohesion. We have revised the section detailing specific examples of global models (GHMs, LSMs, and DGVMs) used within ISIMIP into a more concise summary paragraph (Lines 56 – 70 in the new manuscript). The Paragraph is presented here for quick reference

Line 56:

Within the ISIMIP ensemble (Table 1), various models exemplify different categories. For instance, GHMs like (Burek et al., 2020) focus on water quantity assessment across sectors (Becher et al., 2024; Palazzo et al., 2024), H08 (Hanasaki et al., 2018) maps water abstractions and availability, PCR-GLOBWB (Sutanudjaja et al., 2018) simulates the terrestrial water cycle including human influences (Becher et al., 2024; Burek et al., 2020; Palazzo et al., 2024), The Variable Infiltration Capacity (VIC) assesses human impacts on water resources (Liang et al., 1994), and WaterGAP (Müller Schmied et al., 2021) simulates runoff, recharge, and streamflow considering various storages. LSMs such as CLM, CLASSIC, JULES, and MATSIRO (Best et al., 2011; Lawrence et al., 2019; Melton et al., 2020; Takata et al., 2003) simulate broader land surface processes including energy, water, carbon fluxes, and vegetation dynamics, often incorporating hydrological components. DGVMs like LPJmL (Sitch et al., 2003) simulate vegetation dynamics and surface water balance, including human influences like irrigation. Despite differing primary objectives, these models are often used collectively in ensemble studies to assess CC impacts on various hydrological aspects like groundwater recharge (Reinecke et al., 2021), river flow (Gudmundsson et al., 2021) and soil moisture (Porkka et al., 2024), river ecological functioning (Thompson et al., 2021), droughts (Kew et al., 2021; Pokhrel et al., 2021), and floods (Dottori et al., 2018; Tabari et al., 2021; Zhou et al., 2023), both globally and regionally.

- The comparison of SWAT+ ET with the GLEAM dataset lacks a clear description of the time range. It is essential to specify the exact time period and resolution used for the comparison

Thanks for pointing out the need for clarification regarding the time range and resolution used in the evapotranspiration (ET) comparison. Upon a closer look, we have updated the reference for GLEAM 4 (Line 173) as the final paper has since been published in Nature. We have also included dataset resolution here so that the user does not have to read the referenced manuscript to know this.

We have also made a correction in Section 2.3) specifying that evaluation was done with GLEAM v4. We have revised Section 3.1 to explicitly state that the comparison between SWAT+ simulated ET and the GLEAM dataset covers the effective simulation period of 1982-1990 (after the 5-year warm-up)

Line 173:

GLEAM v4 dataset available at 0.1° resolution was used for evaluating ET (Miralles et al., 2025). The datasets require pre-processing to be used by the SWAT+ model.

Line 263:

A comparison of ET for the effective simulation period (1982 – 1990) shows that the spatial pattern between SWAT+ ET and GLEAM ET is comparable overall.

- The manuscript lacks a comparison analysis for the performance of monthly river discharge with other ISIMIP global hydrological models. The comparison is essential to provide a comprehensive understanding of the performance and limitations of the global SWAT+ model.

Thank you for your suggestion regarding a comparative analysis of monthly river discharge performance against other ISIMIP global hydrological models. We acknowledge the value of model intercomparison exercises like ISIMIP for contextualizing model performance. However, a direct, quantitative performance comparison with the existing ISIMIP model ensemble was not a primary objective of this particular study. Our main goals were to: (1) Develop the high-resolution (2km) global SWAT+ model setup, which is novel for SWAT+ at this scale, (2) establish the reproducible CoSWAT framework to manage the associated data and computational challenges, and (3) provide an initial, uncalibrated baseline evaluation of this high-resolution model's performance using standard metrics and datasets.

Furthermore, conducting a direct, station-by-station performance comparison between our 2km model and the standard ISIMIP model outputs (typically available at 0.5-degree resolution) presents significant conceptual and technical challenges. The fundamental difference in spatial scale means that our model explicitly represents fine-scale landscape heterogeneity, hydrological processes, and potentially smaller river systems that are inherently averaged or entirely unresolved within a 0.5-degree grid cell (~55km at the equator). Evaluating both model types against the same point-based gauge data, therefore, does not constitute a like-for-like comparison. For instance, the hydrological response at a gauge used for evaluation might be dominated by processes within a catchment much smaller than a 0.5-degree cell, processes which only the high-resolution model attempts to capture. Directly comparing KGE values, for example, could be misleading as the models are simulating fundamentally different spatial representations of the hydrological system.

While the CoSWAT framework developed here could potentially facilitate such comparisons in the future (e.g., by aggregating SWAT+ outputs or comparing against upscaled observations), our focus in this initial paper was on demonstrating the feasibility and presenting the baseline characteristics of the high-resolution global SWAT+ simulation itself. We have added a note in the Discussion section acknowledging this limitation and suggesting detailed inter-model comparison, accounting for scale differences, as an avenue for future research.

Line 356:

Finally, while placing model performance in the context of established global models like those within the ISIMIP ensemble is valuable, a direct quantitative comparison of river discharge statistics (e.g., KGE) was considered beyond the scope of this initial study and potentially misleading due to fundamental differences in model resolution. Comparing our high-resolution outputs, which capture finer-scale heterogeneity, against typical ISIMIP model outputs (0.5-degree) at specific gauge locations requires careful consideration of scale mismatches. Future work could explore methodologies for robust inter-comparison that account for these scale differences, potentially leveraging the aggregation capabilities of the CoSWAT framework.

- In Table 1, Nr 1 "CWatM" should maintain consistent capitalization with "CWATM" in line 57.

  We thank the reviewer for catching this, we have made the suggested correction in the revised manuscript for all references of CWatM.

- In line 66, the abbreviation "CC" is introduced without its full form. This should be corrected to enhance the clarity of the manuscript.

  Good catch! In the revised manuscript, we have now fully introduced the abbreviation upon mentioning it for the first time in main text (Line 33).

- In line 101, the abbreviation "ORCHIDEE" and "SWBM" is introduced without its full form.

  Due to the need to shorten the model description paragraphs, we have removed direct reference to these two models within main text.

- In line 109, the phrase "with between" in the sentence "Despite the differences in general purposes and focus for model development with between LSMs, GHMs and DGVMs" is grammatically incorrect, please remove "with".

  This has been reviewed as suggested.

- In line 144, in the sentence "increased uncertainty (Sood & Smakhtin, 2015) in model outputs (Sood & Smakhtin, 2015)", the same reference is cited twice in close proximity, which is unnecessary and can be confusing for readers.

  Thanks for pointing this out, we have made the correction.

- In line 211, The sentence "Gleam4 dataset was used for evaluating ET (Miralles et al., 2011, 2024) The datasets require preprocessing to be used by the SWAT+ model." contains a grammatical error. It appears to be a comma splice, where two independent clauses are joined without proper punctuation.

  This sentence has now been updated and reads as follows:

  Line 173:

  GLEAM v4 dataset available at 0.1° resolution was used for evaluating ET (Miralles et al., 2025). The datasets require pre-processing to be used by the SWAT+ model.

- In line 211, "Gleam4 dataset was used for evaluating ET", In line 274, "We also evaluated the ET output against GLEAM v3 dataset". It is unclear whether these refer to the same dataset or different versions of the dataset. For clarity and consistency, the authors should ensure that the dataset names are used accurately and consistently throughout the manuscript. The word "Gleam", the authors should maintain consistency when referring to GLEAM. Please revise the terminology throughout the entire paper accordingly.

  We thank the reviewer for this feedback. We have made correction regarding the inconsistency as clarified under the second comment. We have also revised all references in the manuscript for consistency including proper capitalisation.

- In line 259, mentions "HRUs", a detailed introduction to HRU is needed.

We agree that introducing this SWAT+ concept earlier is needed. We have added a brief explanation of Hydrologic Response Units (HRUs) within the paragraph that introduces the SWAT+ model in the Introduction section, clarifying that HRUs are the basic computational units representing unique land use, soil, and slope characteristics.

Line 90:

SWAT+ (Soil and Water Assessment Tool) is a completely revised version of the original SWAT model (Arnold et al., 2018; Bieger et al., 2017). It performs hydrological simulations at the Hydrologic Response Units (HRU) scale. HRUs represent unique combinations of land use, soil, and slope characteristics within each landscape unit or subbasin. The SWAT+ model can simulate a wide range of processes including surface runoff and infiltration, evapotranspiration...

- In line 274, The sentence "We also evaluated the ET output against GLEAM v3 dataset using maps and sample point difference distribution." is somewhat ambiguous. It is unclear how exactly the evaluation was conducted using "maps and sample point difference distribution." For clarity, the authors should provide more specific details about the methods used for this evaluation.

Thank you for highlighting the ambiguity in our description of the evapotranspiration (ET) evaluation method. We agree that more specific details were needed. We have revised the relevant sentence in the Methodology section (originally line 140) to explicitly describe how the maps and sample point difference distribution were used. Specifically, we now state that the maps were visually compared to assess spatial pattern agreement and that the sample point difference distribution was generated by calculating the difference between SWAT+ and GLEAM ET at quasi-randomly selected global points (shown in Figure 6) and plotting the frequency of these differences to evaluate overall bias and distribution around zero.

**References**

[revised manuscript text omitted]

---

## Author Comment (AC2)

**Responses to Reviewers**

Green is quoted comment; Orange is a response, and Blue is quoted text from manuscript.

**Reviewer 2**

- I think the introduction section needs some work. I found this section to be meandering and in my opinion these many separate paragraphs about the modelling applications are not needed. My recommendation is to squeeze these texts in few paragraphs and then try to elaborate the research gap and SWAT large scale applications as well. If I remember correctly, there were a handful of studies that reported comparing SWAT model in very large scales, are we missing some relevant references here?

    Thank you for the feedback on the introduction and for prompting us to ensure we have included relevant references regarding large-scale SWAT applications. We have condensed the discussion of general GHM applications as requested

    Regarding the specific point on large-scale SWAT studies, our work aims to build upon previous efforts by developing the first high-resolution *global* SWAT+ model. The most relevant precursors in terms of scale are indeed the continental applications. We have cited the key studies in this domain within the introduction (Line 100), specifically:

    - ***SWAT*** : The continental-scale model for Europe by Abbaspour et al. (2015).

    - ***SWAT+:*** The recent continental-scale models developed for Africa (Chawanda et al., 2024; Nkwasa et al., 2024)

    While a limited number of valuable studies have applied SWAT+ to large *river basins* worldwide (in part due to SWAT+ being relatively new), these continental-scale applications represent the largest spatial domains modelled prior to our global study and thus form the most direct context for the research gap our work addresses – the lack of a fully global, high-resolution SWAT+ implementation. We believe these references adequately cover the state-of-the-art in very large-scale SWAT+ modelling relevant to our study's objectives.

- The issue that the authors reported as a potential future improvement with regards to the comparison with SWAT simulated global ET with the GLEAMS dataset are in my opinion somewhat previously known knowledge. Many researchers have reported the issue with GLEAM dataset that the dataset due to partitioning of ET, tends to overestimate transpiration while underestimating soil evaporation. For example see Chen et al. (2022) "Uncertainties in partitioning evapotranspiration by two remote sensing-based models". I also believe that GLEAM v4 is available now, with a higher spatial resolution than the v3. Have the authors considered using this product to do their comparison?

    Thank you for your comment regarding the discussion of the evapotranspiration (ET) comparison and the GLEAM dataset.

    Regarding the potential reasons for discrepancies, we agree that differences between modeled ET and remote sensing products can arise from uncertainties in both the model and the benchmark dataset. We have revised the discussion in Section 4 to reflect this more explicitly. While we maintain that input data limitations (e.g., climate data resolution, lack of lake representation) contribute to some observed differences, we now also acknowledge that inherent characteristics and potential uncertainties within the GLEAM product itself, such as its partitioning methods as you noted, can also play a role in comparison results. We appreciate the reference provided (Chen et al., 2022) and have incorporated this perspective into our revised discussion.

    Line 312:

    For instance, the East African rift valley lake area was all simulated with regular HRUs while implementing lakes would ensure that the land ET and lake ET are not mixed up to improve spatial Pattern (Fig 9). Concurrently, inherent

uncertainties within the GLEAM v4 product itself, potentially related to its algorithms for partitioning ET components such as transpiration vs. soil evaporation, as discussed in studies like Chen et al. (2022), can also influence the comparison results. Thus, there is a need to acknowledge these combined uncertainties when interpreting the evaluation of the ET spatial patterns.

Regarding the GLEAM version, we confirm that GLEAM v4 was indeed used for the comparison in this study. We apologize for any confusion caused by an inconsistency in the manuscript text. We have ensured that the manuscript now consistently refers to GLEAM v4 throughout the methods and results sections (including updates to lines 173 and any other mentions).

- I am missing the details about time range for the ET comparison.

Indeed, we now have highlighted the time range used.

Line 263:

A comparison of ET for the effective simulation period (1982 – 1990) shows that the spatial pattern between SWAT+ ET and GLEAM ET is comparable overall.

- I would also like to know was non-availability of spatial data the sole reason for choosing the ASTER GDEM data over SRTM? I think this is important because SRTM does better representation of mountainous regions than ASTER GDEM, so I would like to see what the authors think about this issue.

Thank you for raising the point about the relative qualities of ASTER GDEM and SRTM, particularly concerning mountainous regions. We acknowledge that at their native resolutions, SRTM data is often cited as having advantages in complex terrain compared to earlier ASTER versions.

However, for our specific application – developing a global hydrological model – the primary limiting factor of SRTM is its incomplete spatial coverage, as it does not extend to latitudes beyond 60°N and 56°S. Since our goal was to model the global landmass (excluding Greenland), the comprehensive spatial extent offered by ASTER GDEM was essential.

Furthermore, while differences in data quality (e.g., noise, artifacts) between the two DEMs are more pronounced at their native ~30-90m resolutions, our global model utilizes these data resampled to a 2km resolution. At this coarser scale, used for watershed delineation and topographic parameterization in a global context, the finer-scale differences between ASTER and SRTM become less significant for capturing the overall large-scale topographic features driving hydrological processes. Therefore, given the necessity of complete spatial coverage for our global domain and the mitigating effect of our 2km working resolution, ASTER GDEM was deemed the most suitable choice for this study. We have added a sentence to the manuscript in Section 2.1.1 to briefly clarify this consideration.

Line 144:

The Advanced Spaceborne Thermal Emission and Reflection Radiometer (ASTER) global DEM (Abrams, 2016) was preferred over the Shuttle Radar Topography Mission (SRTM) global DEM (Farr et al., 2007) primarily due to its more complete global spatial coverage (Fig. 1), which is essential for this study's domain. While potential differences in DEM quality exist between the datasets, particularly in mountainous regions at finer native resolutions, these differences are considered less critical at the 2km resolution used for deriving topographic parameters in this global model setup.

- Could the authors provide any details about the actual runtime for basins of different spatial scale?

Thank you for asking for details on computational runtime. This is indeed an important practical aspect of large-scale modeling. We have added specific examples to the Methodology section (Section 2.2) based on our experience setting up and running the model using the CoSWAT framework on the 64-core HPC environment described in the paper.

We clarified that runtime for both model setup (data processing, watershed delineation, HRU generation, file writing) and simulation execution depends significantly on the specific hardware (CPU cores, clock speed, I/O performance) and the parallel processing configuration employed within the framework

Line 206:

The CoSWAT framework was optimised by iteratively implementing parallel processing wherever possible and feasible. This reduces the time required for data processing and model setup, making large-scale simulations feasible by leveraging High Performance Computing (HPC) environments which often allow highly parallelised workflows (Chawanda et al., 2020). The efficiency gains from parallel processing significantly reduce computation time, though actual runtimes depend heavily on the specific HPC hardware (CPU cores, clock speed, Input/Output (I/O) speed) and parallel configurations used. For context, using the 64-core, 3.00 GHz, 128 GB RAM Linux environment described below (Section 2.3), the CoSWAT framework setup phase (including data preprocessing, watershed delineation, HRU generation, and file writing – not including data download times) required approximately 12 minutes for a moderately sized region such as Save Basin in Africa, and about 1 hour 49 minutes for a large, complex region such as the Nile Basin. Executing a 10-year SWAT+ simulation for very large basins like the Amazon could take over 24 hours, with runtime strongly influenced by the requested output frequency (e.g., daily outputs requiring significantly more time due to I/O demands).

- Line 67 : Explain CC

We now write CC in full upon first mention as suggested.

**References**

Abbaspour, K. C., Rouholahnejad, E., Vaghefi, S., Srinivasan, R., Yang, H., and Kløve, B.: A continental-scale hydrology and water quality model for Europe: Calibration and uncertainty of a high-resolution large-scale SWAT model, J. Hydrol., 524, 733–752, https://doi.org/10.1016/j.jhydrol.2015.03.027, 2015.

Abrams, M.: ASTER GLOBAL DEM VERSION 3, AND NEW ASTER WATER BODY DATASET, Int. Arch. Photogramm. Remote Sens. Spat. Inf. Sci., XLI-B4, 107–110, https://doi.org/10.5194/isprs-archives-XLI-B4-107-2016, 2016.

Chawanda, C. J., George, C., Thiery, W., Griensven, A. V., Tech, J., Arnold, J., and Srinivasan, R.: User-friendly workflows for catchment modelling: Towards reproducible SWAT+ model studies, Environ. Model. Softw., 134, 104812, https://doi.org/10.1016/j.envsoft.2020.104812, 2020.

Chawanda, C. J., Nkwasa, A., Thiery, W., and Van Griensven, A.: Combined impacts of climate and land-use change on future water resources in Africa, Hydrol. Earth Syst. Sci., 28, 117–138, https://doi.org/10.5194/hess-28-117-2024, 2024.

Chen, H., Zhu, G., Shang, S., Qin, W., Zhang, Y., Su, Y., Zhang, K., Zhu, Y., and Xu, C.: Uncertainties in partitioning evapotranspiration by two remote sensing-based models, J. Hydrol., 604, 127223, https://doi.org/10.1016/j.jhydrol.2021.127223, 2022.

Farr, T. G., Rosen, P. A., Caro, E., Crippen, R., Duren, R., Hensley, S., Kobrick, M., Paller, M., Rodriguez, E., Roth, L., Seal, D., Shaffer, S., Shimada, J., Umland, J., Werner, M., Oskin, M., Burbank, D., and Alsdorf, D.: The Shuttle Radar Topography Mission, Rev. Geophys., 45, 2005RG000183, https://doi.org/10.1029/2005RG000183, 2007.

Nkwasa, A., James Chawanda, C., Theresa Nakkazi, M., Tang, T., Eisenreich, S. J., Warner, S., and Van Griensven, A.: One third of African rivers fail to meet the 'good ambient water quality' nutrient targets, Ecol. Indic., 166, 112544, https://doi.org/10.1016/j.ecolind.2024.112544, 2024.

---

## Author Response (AR3)

Dear Editor,

We apologise for the oversight of forgetting to update figure hyperlinks. We have updated all figure numbers and synchronised cros-references throughout the manuscript. We thank you for catching this error.